# *Rosa canina L*. Can Restore Endoplasmic Reticulum Alterations, Protein Trafficking and Membrane Integrity in a Dextran Sulfate Sodium-Induced Inflammatory Bowel Disease Phenotype

**DOI:** 10.3390/nu13020441

**Published:** 2021-01-29

**Authors:** Dalanda Wanes, Mohamad Toutounji, Hichem Sebai, Sandra Rizk, Hassan Y. Naim

**Affiliations:** 1Department of Biochemistry, University of Veterinary Medicine Hannover, Bünteweg 17, 30559 Hannover, Germany; dalanda.wanes@tiho-hannover.de (D.W.); toutounji.m@gmail.com (M.T.); 2Laboratory of Functional Physiology and Valorization of Bioresources, Higher Institute of Biotechnology of Beja, University of Jendouba, Beja 7800, Tunisia; sebaihichem@yahoo.fr; 3Faculty of Sciences of Bizerte, University of Carthage, Zarzouna 7021, Tunisia; 4Department of Natural Sciences, Lebanese American University, Beirut 1102-2801, Lebanon; sandra.rizk@lau.edu.lb

**Keywords:** *Rosa canina* methanol extract, polyphenols, inflammatory bowel disease, endoplasmic reticulum stress, protein trafficking, intestinal proteins, sucrase-isomaltase, dipeptidyl peptidase 4, brush border membrane, lipid rafts

## Abstract

*Rosa canina L*. is a natural polyphenol-rich medicinal plant that exhibits antioxidant and anti-inflammatory activities. Recent in vivo studies have demonstrated that a methanol extract of *Rosa canina L.* (RCME) has reversed an inflammatory bowel disease (IBD)-like phenotype that has been triggered by dextran sulfate sodium (DSS) in mice. In the current study, we investigated the effects of RCME on perturbations of cellular mechanisms induced by DSS-treatment of intestinal Caco-2 cells, including stress response in the endoplasmic reticulum (ER), protein trafficking and sorting as well as lipid rafts integrity and functional capacities of an intestinal enzyme. 6 days post-confluent cells were treated for 24 h with DSS (3%) or simultaneously with DSS (3%) and RCME (100 µg/mL) or exclusively with RCME (100 µg/mL) or not treated. The results obtained demonstrate the ability of RCME to counteract the substantial increase in the expression levels of several ER stress markers in DSS-treated cells. Concomitantly, the delayed trafficking of intestinal membrane glycoproteins sucrase-isomaltase (SI) and dipeptidyl peptidase 4 (DPP4) induced by DSS between the ER and the Golgi has been compromised by RCME. Furthermore, RCME restored the partially impaired polarized sorting of SI and DPP4 to the brush border membrane. An efficient sorting mechanism of SI and DPP4 is tightly associated with intact lipid rafts structures in the *trans*-Golgi network (TGN), which have been distorted by DSS and normalized by RCME. Finally, the enzymatic activities of SI are enhanced in the presence of RCME. Altogether, DSS treatment has triggered ER stress, impaired trafficking and function of membrane glycoproteins and distorted lipid rafts, all of which can be compromised by RCME. These findings indicate that the antioxidants in RCME act at two major sites in Caco-2 cells, the ER and the TGN and are thus capable of maintaining the membrane integrity by correcting the sorting of membrane-associated proteins.

## 1. Introduction

Inflammatory bowel diseases (IBD) are chronic inflammatory disorders of the gastrointestinal tract with two major types: Crohn’s disease and ulcerative colitis. The etiology of IBD implicates genetic, immune and environmental factors that ultimately disturb homeostasis, which implicates a well-coordinated dynamic function of intestinal microbiota and epithelium as well as the immune system. Several studies have shown that the intestinal digestive function and absorption are impaired in IBD [1,2], likely due to inflamed and subsequently altered epithelial morphology [3,4], which result in diverse symptoms, such as diarrhea, abdominal pain and cramping. Intestinal sucrase-isomaltase (SI) is one of those major enzymes that are responsible for the final steps of digestion of disaccharides. Other enzymes in this group are maltase-glucoamylase and lactase-phlorizin hydrolase. Reduced levels of SI at the brush border membrane (BBM) elicit carbohydrate malabsorption and associated symptoms, such as diarrhea, bloating, abdominal pain and cramping. The SI-related malabsorption can be primary as in genetically-determined SI deficiency or irritable bowel syndrome (IBS) or secondary, as in IBD, celiac disease and several functional gastrointestinal disorders. SI is a type II membrane glycoprotein that is trafficked with high fidelity along the secretory pathway to the apical membrane of intestinal cells; the polarized sorting of SI occurs via O-linked glycans that mediate its association with lipid rafts (LRs) or detergent-resistant membranes (DRMs) [5].

The correct folding and maturation of intestinal proteins require a robust endoplasmic reticulum (ER) function. The accumulation of misfolded proteins in the ER of intestinal epithelial cells can trigger ER stress, thereby activating the unfolded protein response (UPR) and consequently affecting ER homeostasis. The disturbance of ER homeostasis or the disruption of UPR in the gut may lead to IBD through the upregulation of proinflammatory cytokines, the impairment of mucosal barrier function and/or increasing apoptosis in intestinal epithelial cells [4]. In this aspect, several factors have been suggested to contribute to the pathogenesis of IBD, such as protein misfolding in the ER. Previous studies confirmed that the levels of ER stress markers were significantly increased in intestinal epithelium of patients with active IBD [6,7].

Even though different experimental models have been used to mimic IBD conditions, the dextran sodium sulfate (DSS) is the most widely used in vivo murine model [8]. However, in vitro culture models are indeed an important tool to unravel and investigate cellular and molecular mechanisms involved in disease pathogenesis. Previous studies in our laboratory have confirmed that Caco-2 cells are an adequate model to study the epithelial disruption upon DSS treatment [9].

The current therapies available for IBD are mainly corticosteroids, anti-inflammatory in addition to immunosuppressive and biological agents [10]. However, these medications can present serious side effects and high risk of complications, which limit their use [11,12,13]. Moreover, due to the high relapse rate of IBD, it is critical to find alternative sources of treatment. Several phenolic compounds have been reported to modulate or improve the inflammatory response in IBD [14]. In a murine DSS-induced colitis model, the administration of polyphenols induced a downregulation of inflammatory markers such as the monocyte chemoattractant protein 1 (MCP1) and tumor necrosis factor α (TNFα) [15]. Similarly, in a recent study, the administration of a crude extract of *Rosa canina* flowers to DSS-intoxicated mice protected against DSS-induced oxidative stress in the colon. Moreover, the DSS- induced histopathological damage was alleviated by *Rosa canina* extract [16]. *Rosa canina L*., a small shrub of Rosa genus belonging to the Rosaceae family [17], is widespread in the temperate and subtropical zones of the Northern hemisphere [18]. *Rosa canina L.* possesses therapeutic activities against various inflammatory disorders such as rheumatism, arthritis and gout, in addition to other diseases like cold and influenza [19,20]. *Rosa canina L.* was used as a treatment for gastrointestinal tract diseases such as gastric ulcers [21] and diarrhea [22]. The biological activities of *Rosa canina L.* are attributed to the high levels of bioactive molecules such as vitamin C, carotenoids, anthocyanins, and phenolic acids [16,23].

The aim of this study was to investigate the effect of *Rosa canina* methanol extract (RCME) on DSS-induced disruption of ER homeostasis and its downstream effects on protein trafficking and sorting of intestinal SI and DPP4, in addition to LRs in intestinal Caco-2 cells.

## 2. Materials and Methods

### 2.1. Cell Culture and Treatment

The human Colon carcinoma Caco-2 cells were purchased from the German Collection of Microorganisms and Cell Cultures (DSMZ, Braunschweig, Germany). Cells were cultured in Dulbecco’s Modified Eagle Medium (DMEM, Sigma-Aldrich, Darmstadt, Germany) supplemented with 10% FCS (Sigma, Darmstadt, Germany), l-glutamine (4 mM), pyruvic acid (1 mM), D-glucose (4.5 mg/mL) and penicillin/streptomycin (100 U/mL). Cells were maintained at 37 °C in a humidified atmosphere (5% CO2 and 95% air). 6 days post-confluent cells were treated for 24 h with DSS (3%, 500 kDa; MP biomedicals GmbH, Eschwege, Germany) or simultaneously with DSS (3%) and RCME (100 µg/mL) or exclusively with RCME (100 µg/mL). The control employed 7 days post-confluent cells without any treatment. The treated or non-treated cells were subjected to cytotoxicity assay as described below (Section 2.3).

### 2.2. RCME Preparation

RCME was prepared as described before [16]. Briefly, the powder from dried *Rosa canina* flowers was macerated in 80% methanol (*w*/*v*) and dried afterwards in rotor-evaporator followed by freeze drying.

### 2.3. Cytotoxicity Assay

The measurement of cell death was conducted with lactate dehydrogenase (LDH) assay using CytoTox-ONE™ Promega kit (Promega Corporation, Fitchburg, WI, USA). LDH is a soluble cytosolic enzyme released upon cell membrane damage indicating cell death.

Briefly, the supernatant of post-confluent Caco-2 cells, previously treated with DSS (3%) or RCME (50, 100, 500 and 1000 µg/mL) (see Section 2.1), was mixed with the kit reagent and incubated for 10 min in the dark. The negative control consists of medium only, the control refers to the supernatant from non-treated cells and the positive control refers to the supernatant from cells lysed with 1% Triton X-100. The level of LDH release was measured at an excitation wavelength of 560 nm and an emission wavelength of 590 nm, according to the manufacturer’s instructions.

### 2.4. Total RNA Isolation and Reverse Transcription

Six days post confluent cells were treated for 24 h with DSS and/or RCME as described above. Total RNA was extracted with TRIzol reagent (Thermo Fisher Scientific, Waltham, MA, USA) followed by the cDNA synthesis with poly (dT) primer using Super Script III Reverse Transcriptase (Invitrogen, Carlsbad, CA, USA). Both procedures were performed according to the manufacturers’ protocols. The resulting cDNA was used for semi-quantitative RT-PCR experiments to assess the expression levels of the ER stress markers, activation transcription factor 4 (ATF4), immunoglobulin-binding protein (BiP), C/EBP homologous protein (CHOP), ERAD-enhancing α-mannosidase-like proteins (EDEM), glucose regulated protein 94 (Grp94) and X-box binding protein (XBP1) using the primers listed in Table 1 and according to the experimental protocol described before [24]. Glyceraldehyde 3-phosphate dehydrogenase (GAPDH) was used as a housekeeping gene.

### 2.5. Biosynthetic Labeling, Immunoprecipitation and Deglycosylation of Sucrase-Isomaltase (SI) and Dipeptidyl Peptidase 4 (DDP4)

Seven days post-confluent Caco-2 cells from the different treatment groups were starved for 2 h in methionine-free medium and then biosynthetically labelled with 100 µCi (35S) methionine for different time points. The labeling time points were chosen depending on the transport kinetics of the two proteins being investigated, namely SI and dipeptidyl peptidase 4 (DPP4). For the analysis of SI the cells were continuously labeled for 2 h and 6 h and for DPP4 0.5 h and 2 h. After labelling the cells were washed twice with ice cold phosphate-buffered saline (PBS) and solubilized for 1 h at 4 °C in a lysis buffer containing 25 mM Tris-HCl, pH 8.0, 50 mM NaCl, 0.5% Triton X-100, 0.5% sodium deoxycholate and a mixture of protease inhibitors. The cellular lysates were centrifuged at 17,000× *g* for 20 min at 4 °C and the cellular extracts were retained and immunoprecipitated with monoclonal antibodies directed against SI (mAbs HBB2/614/88 and HSI2) and DPP4 (mAb HBB 3/775/42) [25,26]. The antibodies were generously provided by Drs. E.E. Sterchi and H.-P. Hauri, formerly at the University of Bern and University of Basel, Switzerland, and Dr. B. Nichols, Baylor College of Medicine, Houston, USA.

The immunoprecipitated SI and DPP4 were thereafter treated with endo-β-*N*-acetylglucosaminidase H (Endo H) as previously described [27] to discriminate between the mannose-rich glycosylated from the complex glycosylated protein forms. Briefly, the immunoprecipitated proteins were eluted from protein-A Sepharose beads by boiling at 95 °C for 10 min in denaturing buffer (100 mM Tris-HCl, pH 7.5, 1% 2-mercaptoethanol and 1% SDS). Thereafter, the eluted proteins were incubated with Endo H (2.5 mU; Sigma Aldrich, Darmstadt, Germany) for 1 h at 37 °C. Treated or non-treated samples were finally analyzed by SDS-PAGE (described in Section 2.8.).

### 2.6. Brush Border Membrane (BBM) Preparation

BBM isolation utilized the divalent cation-based separation procedure using CaCl_2_ and following the initial protocol by Schmitz et al. [28] and the modification by Sterchi and Woodley [29]. Here, the Caco-2 cells that have been treated with DSS, DSS and RCME, RCME alone or not treated were first homogenized in a buffer containing 12 mM Tris-HCl, pH 7.0 and 300 mM mannitol (referred to as homogenization buffer) using a Potter-Elvehjem homogenizer. The homogenates were cooled and subjected to several ultrasonic pulses on ice (3 times, 10 s each, 15 microns). Cell debris were removed from the homogenates by centrifugation at 5000× *g* for 15 min at 4 °C. CaCl_2_ was added to the cleared homogenates from a 100 mM stock solution to a final concentration of 10 mM. The treated homogenates were left for 30 min at 4 °C with gentle rotation followed by 20 min centrifugation at 5000× *g* to yield a pellet (P1), representing the intracellular and basolateral membranes, and a supernatant (S1). Thereafter, S1 was centrifuged at 25,000× *g* for 30 min at 4 °C to yield the pellet (P2) representing BBM. P1 and P2 were then resuspended in the homogenization buffer and used for further biochemical analysis.

### 2.7. Preparation of Detergent-Resistant Membranes

The isolation of cholesterol-sphingolipids enriched DRMs or LRs was performed on a discontinuous sucrose density gradient [30]. Briefly, treated or non-treated Caco-2 cells were solubilized at 4 °C for 2 h with 1% Triton X-100 prepared in phosphate-buffered saline (pH 7.4) (*w*/*v*). The cellular extracts were then centrifuged at 5000× *g* for 20 min at 4 °C to remove cell debris. The supernatant (denoted lysates) was loaded on a sucrose gradient composed of 1 mL 80% *w*/*v* sucrose, 1 ml lysates in 40% *w*/*v* sucrose, 7 mL 30% *w*/*v* sucrose and 1 mL 5% *w*/*v* sucrose and subjected to ultracentrifugation at 100,000× *g* for 18 h at 4 °C using an SW40 rotor (Beckman Coulter, Mississauga, ON, Canada). Ten fractions (1 mL each) were collected from the top to the bottom of the gradient. The floating top 3 fractions are considered as LRs and the 3 last bottom fractions correspond to the non-lipid raft fractions (non-LRs). The collected fractions were subjected to SDS-PAGE to examine the distribution of the LR marker, flotillin-2 (FLOT2) (Sigma, Munich, Germany).

A quick extraction method was also used to separate LRs and non-LRs or detergent-soluble membranes (DSMs) [31]. Briefly, a similar solubilization procedure with 1% Triton X-100 was performed as described above, with the modification that the cellular extracts were ultracentrifuged at 100,000× *g* for 1 h at 4 °C. The collected pellet contained the LRs and the supernatant contained the non-LRs. The LRs and the non-LRs were analyzed by SDS-PAGE and Western blotting to assess the levels of SI in these two fractions.

### 2.8. SDS-PAGE, Western Blotting and Fluorography

Protein samples were resolved by SDS-PAGE on 6% or 8% polyacrylamide slab gels according to Laemmli [32] under reducing conditions using 10 mM dithiothreitol (DTT). The gels were blotted onto PVDF membranes (240 mA, 1.5 h). The membrane was blocked in 5% skimmed milk in PBS and 0.1% Tween 20 (PBST). The membranes were treated with either one of the following primary antibodies: HBB3/705/60 for SI (1 µg/µL), mAb HBB 3/775/42 for DPP4 (1 µg/µL) [25,26] and anti-flotillin-2 B-6 (0.2 µg/µL; Santa Cruz Biotechnology, Dallas, TX, USA) for 1 h at RT. The blots were washed 3 times with PBST and subjected to further treatment with the secondary antibody, anti-mouse IgG conjugated to horseradish peroxidase (0.4 µg/µL; Thermo Fisher Scientific). All the antibodies were used at a dilution of 1:5000. The protein bands were visualized via enhanced chemiluminescent peroxidase substrate and documented with a ChemiDoc MP™ Touch Imaging System (Bio-Rad, Munich, Germany).

Radioactively-labelled SI and DPP4 were analyzed on 6% gels and the protein bands visualized by fluorography essentially as described before [33] except that 1 M sodium salicylate was used to soak the gel.

### 2.9. Enzymatic Assay

The enzymatic activities of SI, in cellular homogenates and detergent extracts, were measured according to Dahlqvist [34] using sucrose as a substrate. Briefly, the cellular lysates were incubated with the sucrose substrate (150 mM, pH 6.25) for 1 h at 37 °C. Afterwards, glucose oxidase-peroxidase mono-reagent (GOD-PAP) was added to the mixture, incubated for 20 min at 37 °C and the optical density of the released glucose was measured at 492 nm.

### 2.10. Statistical Analysis

The results are presented as mean ± standard error of mean (SEM) of at least three independent experiments. Two-way ANOVA followed by Tukey’s multiple test was used for the comparison between the different groups. GraphPad Prism 8.0.1 (244) software (GraphPad Software, San Diego, CA, USA) was used for the calculations. *p*-values < 0.05 were considered statistically significant.

## 3. Results

The human colon adenocarcinoma cell line Caco-2 has been largely used as an intestinal epithelial cell model due to the ability of these cells to spontaneously differentiate into a polarized monolayer epithelium with typical characteristics of mature absorptive enterocytes [35]. Recently, we confirmed the suitability of Caco-2 cells as a model to investigate alterations in membrane trafficking upon DSS-induced ER stress [9]. In this study, we investigated the potential of RCME in the restoration of impaired intracellular trafficking and rescue of functionality of intestinal proteins in DSS-treated Caco-2 cells.

### 3.1. DSS-Induced ER Stress in Caco-2 Cells Is Corrected by RCME

Prior to analyzing the effects of RCME and DSS on protein trafficking and activity, we asked whether RCME at different concentrations (50, 100, 500 and 1000 µg/mL) and DSS (3%) elicit cytotoxic effects on Caco-2 cells. Figure 1 shows that no significant increase of LDH release was observed in treated versus non-treated Caco-2 cells indicating that neither RCME nor DSS exhibit cytotoxic effects on Caco-2 cells at the concentrations used.

It has been shown that the ER stress is implicated in the development and perpetuation of IBD [36] via activation of the ER stress inducers IRE1α, PERK and ATF6 and upregulation of several ER-stress sensors [37,38,39]. In line with this mechanism, our data demonstrate that the ER sensors CHOP, ATF4, BiP, Grp94, XBP1s and EDEM were significantly upregulated after DSS treatment of Caco-2 cells. The increase in the expression levels varied between one sensor and the other, with ATF4 and EDEM being mostly upregulated (>16 fold), followed by CHOP and GRP94 (>13 fold) and finally XBP1 (around 2.5 fold) and BiP (around 1.5 fold). Having determined the effects of DSS on ER stress factors, we addressed the function of RCME and its potential role in restoring ER homeostasis. As shown in Figure 2, RCME inhibited completely or drastically the effect of DSS, likely due to the polyphenolic content of this extract.

### 3.2. DSS-Induced Delayed Trafficking of Intestinal Proteins from the ER to the Golgi Apparatus Is Restored by RCME

Since RCME treatment has restored the levels of the ER stress proteins back to normal we asked whether this extract would also abolish potential effects of DSS on the trafficking of membrane glycoproteins, which are usually the targets of ER stress proteins. One of the proteins studied here is SI, the major intestinal disaccharidase implicated in the hydrolysis of a wide spectrum of glycosidic linkages in carbohydrates. We have previously shown that the intracellular processing of SI is affected in DSS-treated Caco-2 cells due to its delayed exit from the ER to the Golgi [9]. Therefore, we first confirmed the trafficking delay of SI in biosynthetically labelled DSS-treated Caco-2 cells and then determined whether RCME compromises the DSS effect. As shown in Figure 3A, SI was detected in all experimental samples as a mannose-rich 210 kDa protein band at 2 h of pulse, by virtue of its complete sensitivity towards endo H that cleaves N-linked mannose-rich glycosylated ER-located glycoproteins. At 6 h of labelling, maturation of mannose-rich SI to a complex glycosylated endo H-resistant 245 kDa occurred in all samples albeit to variable proportions. To compare the proportions of the mature versus the mannose-rich forms of SI, we measured the labelling intensities of their endo H-treated forms (i.e., 245 kDa versus 185 kDa), since this approach results in a better separation of the protein bands allowing a more precise and easier comparison of the corresponding intensities. Thus, the ratios of the complex mature SI to the mannose-rich species in the wild type and the DSS-treated samples were 4.7 ± 0.56 versus 1.7 ± 0.37 concomitant with a substantially reduced maturation of SI in the DSS-treated samples and delayed trafficking to and processing in the Golgi. Markedly, the ratio increased upon treatment with RCME to 3.3 ± 0.54 compatible with a partial restoration of the maturation kinetics and trafficking rate of SI. It is worth noting that cells treated with RCME revealed similar maturation pattern of SI as that of non-treated cells as governed by their similar ratios of complex glycosylated versus mannose-rich biosynthetic forms.

In addition to SI, we examined the biosynthesis and processing of DDP4, another apically-located membrane glycoprotein, to determine whether the effects of RCME can be also reproduced with another protein that is trafficked along the secretory pathway. Since DPP4 exhibits faster intracellular trafficking kinetics than SI, we chose different time points for the biosynthetic labelling [25]. After 30 min of pulse, the majority of de novo synthesized DPP4 appeared as an endo H-sensitive mannose-rich protein (Figure 3B, all 4 experimental samples), whereas after 2 h of pulse, most of DPP4 was converted to the 124 kDa mature endo H-resistant form. A change in the biosynthetic forms was not observed in the DSS-treated sample. However, the ratio of the mature complex glycosylated DPP4 to its mannose-rich counterpart was substantially reduced to 4-fold in the DSS-treated versus non-treated cells (from 5.5 ± 1.46 to 1.3 ± 0.26). Remarkably and in a fashion similar to SI, RCME counteracted the DSS effect by increasing the complex to mannose-rich ratio to 2.4 ± 0.42 indicating that these extracts have at least partially restored the trafficking of DPP4.

### 3.3. The Effect of DSS on the Sorting of Intestinal Proteins to the Brush Border Membrane Is Compromised by RCME

The DSS treatment induce a delay in protein trafficking of SI and DPP4, nevertheless appreciable proportions of these proteins, were able to exit the ER and converted to the complex mature form. Given that DSS has induced delay in the trafficking of these proteins from the ER to the Golgi that was partially compromised by RCME, we addressed next the potential role of this extract in restoring the trafficking and sorting of these proteins to the apical membrane that have been altered by DSS. For this purpose, cellular fractionations of Caco-2 cells were performed to separate the BBM (P2 fraction) from intracellular and basolateral membranes (P1 fraction). The enrichment factors of P1 and P2 were normalized by comparing the band intensities of SI and DPP4 in these fractions to that in the total cellular homogenate (H). Figure 4A demonstrates that the enrichment of SI in BBM decreased in DSS-treated Caco-2 to 1.92-fold as compared to 5.5-fold in the non-treated cells and that of DPP4 from 8-fold to 1-fold (Figure 4B). Strikingly, addition of RCME to the DSS-treated cells did not only compromise the DSS effect and restore the expression of SI in the BBM to normal, but rather elevated these levels further to 7.7-fold. The SI levels in the BBM was even more substantial, almost 9-fold, when the cells were treated with RCME alone (Figure 4C).

RCME has also restored the trafficking of DDP4 to BBM (Figure 4B,C), albeit to an overall lesser extent than SI. In fact, RCME compromised the effects of DSS by elevating the enrichment level of DPP4 in DSS-treated cells from 1-fold to almost 5 when RCME was added. Treatment of the cells with RCME alone was not associated with a further increase in the enrichment factor versus the non-treated cells as has been shown for SI. Despite the positive effect of RCME on either protein, the variation in the protein levels retained in the BBM after RCME treatment reflect the difference in the targeting pathways of SI and DPP4. While SI is directly targeted from the *trans*-Golgi network (TGN) to the apical membrane with high fidelity, DPP4 is transported by transcytosis via the basolateral membrane to the apical membrane. It can be postulated that DSS affects both routes rendering their restoration and subsequently the trafficking of transcytosed DPP4 less efficient than that of directly targeted SI leading to reduced levels of DPP4 in BBM.

### 3.4. The Distribution of Flotillin 2 in Lipid Rafts Is Restored by RCME

The final step along the secretory pathways of SI and DPP4, which is the sorting to the apical membrane, is mediated via cholesterol- and sphingolipids-rich microdomains or LRs [5,40]. LRs are detergent-insoluble membranes which are resistant to non-ionic detergents such as Triton X-100 and can be retained in the floating fraction on sucrose density gradients due to their low buoyant density. One of the approaches to identify LRs in cellular preparations is to assess the expression levels of FLOT2, a frequently used LRs marker [41], in these preparations. Figure 5 demonstrates that under normal control conditions, FLOT2 was primarily abundant in the 3 top floating fractions and to a very little extent in the fractions 4–7. In DSS-treated cells, a different distribution pattern of FLOT2 was obtained. In addition to the floating fractions (1–3), FLOT2 was also identified in the Triton X-100 soluble non-raft fractions (8–10), concomitant with a reduction in the overall cellular concentration of LRs. Moreover, there was a distinct reduction in the intensity of FLOT2 in fraction 1, which suggests a change in the LRs composition. When DSS-treated cells were subjected to RCME, FLOT2 disappeared from the bottom fractions representing the Triton X-100 soluble cellular lysates. Interestingly, however, is the persistence of the lower FLOT2 intensity in fraction 1 as in the DSS-treated cells compatible with an incomplete restoration of the LRs back to normal. Finally, RCME treatment alone revealed essentially a similar FLOT2 pattern as in the non-treated controls with similar distribution of FLOT2.

### 3.5. The Enzymatic Activities of Sucrase-Isomaltase Are Substantially Increased in RCME-Treated Cells

LRs, N- and O-glycosylation are implicated in regulating the function of SI in intestinal Caco-2 cells [42]. Thus, the activities of sucrase and isomaltase increase substantially in LRs when N- and O-glycosylation patterns are intact and disruption of LRs reduces substantially these activities. DSS triggers delayed maturation of SI, yet the maturation pattern is essentially similar to that of the control non-treated cells as assessed by endo H-treatments. Given that LRs in DSS-treated cells are distorted, we addressed the relevance of this distortion to the function of SI and measured its specific enzymatic activity after DSS and/or RCME treatments of Caco-2 cells. As shown in Figure 6A the overall specific activity of SI in total cellular homogenate was significantly decreased (>50%, *p* < 0.05) after DSS treatment. Remarkably, incubation of DSS-treated cells with RCME or control cells with RCME did not only restore the sucrase activity but rather increased it significantly by almost 2-fold.

Our data support the notion that reduction in total cellular LRs as assessed by FLOT2 distribution leads to a reduced association of SI with LRs, which is reflected by the reduced SI activities in LRs versus non-LRs (Figure 6B). Incubation of RCME with DSS-treated Caco-2 cells reversed the LRs versus non-LRs proportions and resulted in an increase in the SI activity retained in LRs.

## 4. Discussion

Differentiated Caco-2 cells are polarized epithelial cells that express intestinal carbohydrases and peptidases and exhibit morphological and functional properties typical for absorptive enterocytes [35]. Treatment of Caco-2 cells with DSS, which induces IBD in mice, leads to significant morphological alterations comprising decreased transepithelial resistance, disruption of tight junctions and impaired permeability of the epithelial barrier. Similar cellular phenomena occur also in IBD, rendering Caco-2 cells a useful experimental model to investigate membrane and protein transport and intestinal function upon DSS treatment. In this manuscript we have examined the potential of a methanol extract of *Rosa canina L*. in restoring the cellular hemostasis by reversing the effects induced by DSS that comprise ER stress, impaired protein trafficking and sorting via LRs and enzyme function. Our data show that the first site of action of RCME is the early secretory pathway, since this extract has been capable of normalizing the transcriptional levels of UPR-related protein markers XBP1s, ATF4, CHOP, EDEM and GRP78 (BiP), which have been elevated by DSS. The decreased expression of ER-stress markers may be attributed to the polyphenolic composition of RCME and in particular quercetin [16]. In fact, in previous reports, quercetin has been shown to restore ER homeostasis in endothelial cells under stress conditions, by reducing the expression of BiP, CHOP and caspase-3 [43]. Several other natural compounds have also been shown to modulate and inhibit ER-stress [44,45].

A growing body of evidence suggests that altered redox status, the changes in calcium release from the ER and the activation of inflammation markers, such as the nuclear factor-κB (NF-κB), all can disrupt ER homeostasis and lead to inflammation [46]. Given that the generation of reactive oxygen species (ROS) can trigger ER-stress, using antioxidant molecules to inhibit oxidative stress presents a promising therapeutic approach. The antioxidant properties of RCME have been recently reported, whereby oral administration of RCME in mice revealed protective effects against DSS-induced oxidative stress and the overload of ROS [16]. Interestingly, the levels of BiP were not only restored to normal, but decreased substantially upon treatment with RCME either alone or in combination with DSS treatment. BiP is a heat shock protein (HSP) that is activated by ATP and considered as a primary sensor in ER stress activation [47]. Phenolic compounds are capable of stimulating the expression of chaperones, such as HSP70 [48] and can also repress this expression in a dose-dependent manner [49]. In tumors, the reduction in the expression of HSPs by phenolic compounds is even beneficial since it renders these cells vulnerable [50]. Along similar lines is the inhibition of HSP70 that could prevent the increase of proinflammatory mediators levels [51].

The ER is the organelle in which protein folding [52], acquisition of transport competence, and vesicular budding occur and hence it regulates the life cycle of membrane, secretory, or lysosomal proteins. Substantial overexpression of ER stress factors as demonstrated here would be expected to modulate many of these ER-based events. In fact, concomitant with the overexpression of several stress proteins DSS induced a substantial delay in the trafficking of SI and DPP4, two intestinal brush border hydrolases, from the ER to the Golgi apparatus. Interestingly, this delay has been also compromised by RCME as shown by the restoration of the maturation patterns of these two glycoproteins. These effects can be in all likelihood attributed to the antioxidative effects of polyphenols and flavonoids in RCME that compromised the expression levels of ER chaperones implicated in the folding and transport-competence of SI and DPP4, such as BiP (GRP78) [53] and GRP94 (data not shown). Particularly GRP78, the major regulator of UPR, is a stress sensor that regulates the redox status in the ER and modulates antioxidant activities of glutathione and NAD(P)H:quinone oxidoreductase [54,55].

In intestinal epithelial cells, SI and DPP4 are apically sorted to the brush border membrane with high fidelity [5,56]. These proteins are major enzymes responsible for the digestion of different dietary carbohydrates and peptides and their reduced expression at the cell surface, for instance due to impaired trafficking and missorting, results in intestinal malfunction. In IBD patients, DSS-induced colitis animal models, as well as DSS-treated Caco-2 cells, the cell surface expression and activity of various intestinal hydrolases such as SI and DPP4 is altered [9,57,58]. The partial missorting of SI and DPP4 into the BBM and accumulation in intracellular membranes in DSS-treated Caco-2 cells could be corrected by RCME. This corrective action in late secretory pathway between the Golgi and the cell surface is due to the normalization of LRs that are directly implicated in the sorting of SI and DPP4. The disruption of LRs distribution induced by DSS is mainly attributed to the decrease of cholesterol level in LRs [9]. This result is also in line with pervious observations in which we have demonstrated a subtle reduction in the intensity of FLOT2 in the top fraction of LRs upon treatment of fibroblasts with N-butyl-deoxynojirimycin (known as miglustat), which inhibits sphingolipid synthesis and thus alters the composition of LRs in these cells [59]. The effect of RCME is likely linked to its phenolic composition, specifically luteolin [60], which has the capacity to inhibit lipid peroxidation that occurs in experimental colitis induced by DSS [16]. Recent study demonstrated that resveratrol, a polyphenol compound, impacted positively LRs, as assessed by an increased expression of FLOT2 in sucrose gradients and elevation of fatty acids levels [61]. Another site of action of RCME that could directly restore cholesterol levels is the ER. In fact, the ER is the primary site of the biosynthesis of cholesterol and lipids and GRP78, which is affected by DSS and RCME, is a major modulator of lipid metabolism in the ER [62]. The distorted LRs in DSS-treated cells as assessed by FLOT2 distribution can be therefore restored by RCME at the early level of ER compromising thus the impaired trafficking and sorting of the LRs-associated SI and DPP4 [5,40,63,64].

Finally, the capacities of RCME in acting at the level of the ER to ultimately correct membrane and protein trafficking via LRs is also beneficial to the digestive function of SI (and probably also DPP4). The overall enzymatic activity of SI in LRs is 3-fold higher than its counterpart in non-LRs. A direct relation between LRs association and the enzymatic activities has been demonstrated before in Caco-2 cells. In fact, the activity of SI was substantially elevated when it was associated with LRs and was almost 3-fold more active as compared to SI in non-LRs [42]. When the LRs were disrupted by treatment at 37 °C, the activity of SI declined substantially. Therefore, restoration of LRs by RCME is directly linked to an increase of SI activity back to normal.

Altogether, DSS treatment has induced ER stress, affected trafficking and function of membrane glycoproteins and LRs. Noteworthy, an increased expression of GRP78 and members of UPR, which activate the pro-inflammatory cytokine TNF-α [35,36] that is directly implicated in the pathogenesis of IBD, can be reversed by RCME proposing this mixture to be considered in IBD therapies.

## Figures and Tables

**Figure 1 nutrients-13-00441-f001:**
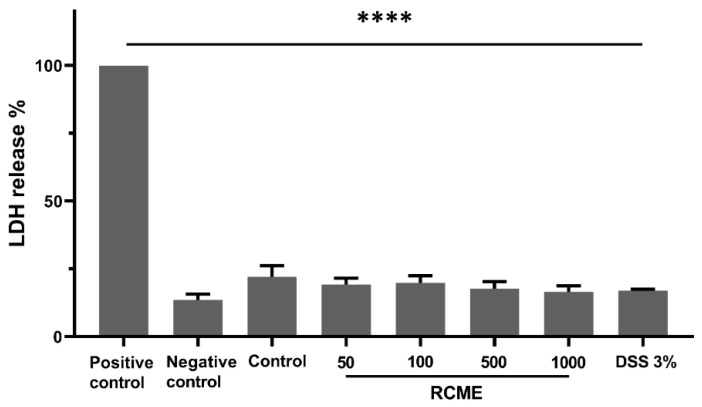
Caco-2 cells viability was not affected upon *Rosa canina* methanol extract (RCME)- or DSS-treatment. Supernatants from Caco-2 cells incubated for 24 h with only medium (Control), different concentrations of RCME (50, 100, 500 or 1000 µg/mL) (RCME), 3% DSS or lysed with Triton X-100 (positive control) were used for measuring LDH level. The negative control refers to medium without cells. Tukey’s multiple test, **** *p* < 0.0001 versus DSS-treated group, S.E.M., *n* = 3.

**Figure 2 nutrients-13-00441-f002:**
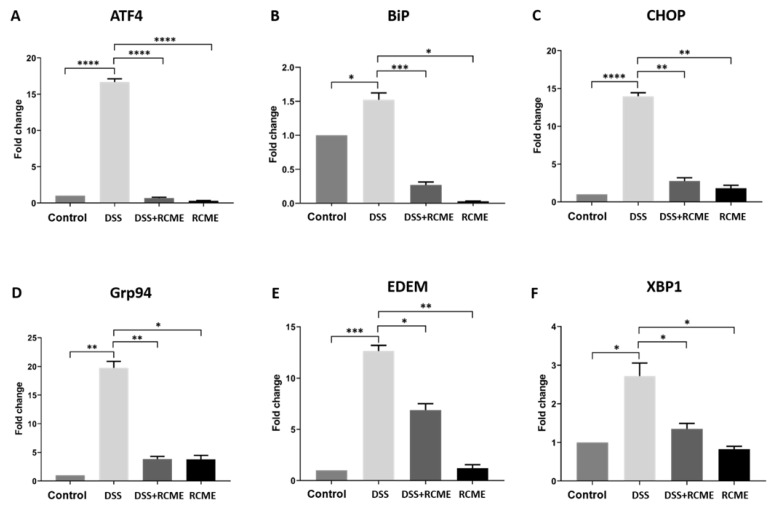
*Rosa canina* methanol extract (RCME) inhibits dextran sulfate sodium (DSS)-induced ER. The expression of ER-stress markers, (**A**) activation transcription factor 4, (**B**) immunoglobulin-binding protein (BiP), (**C**) C/EBP homologous protein (CHOP), (**D**) ERAD-enhancing α-mannosidase-like proteins (EDEM), (**E**) glucose regulated protein 94 (Grp94) and (**F**) X-box binding protein (XBP1) was significantly reduced in RCME-treated groups. Tukey’s multiple test, * *p* < 0.05, ** *p* < 0.01, *** *p* < 0.001, **** *p* < 0.0001 versus DSS-treated group, S.E.M., *n* = 3.

**Figure 3 nutrients-13-00441-f003:**
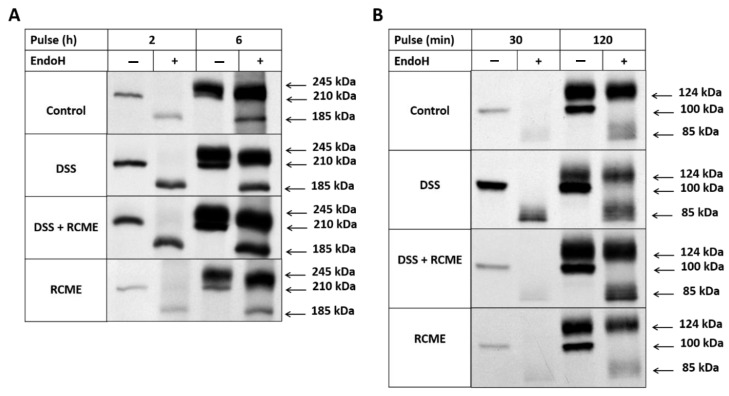
*Rosa canina* methanol extract (RCME) treatment attenuates dextran sulfate sodium (DSS)-induced delayed trafficking of sucrase-isomaltase (SI) and dipeptidyl peptidase 4 (DPP4). Caco-2 cells were treated or non-treated with DSS, DSS and RCME or RCME. 24 h post-treatment, the cells were continuously labelled with [35S] methionine for (2 and 6 h) or for (30 and 120 min) respectively for SI (**A**) and DPP4 (**B**) analysis. The cells were lysed, immunoprecipitated, treated with endo-β-N-acetylglucosaminidase H (endo H) and then subjected to SDS-PAGE followed by fluorography. The figure shown is representative of three different experiments.

**Figure 4 nutrients-13-00441-f004:**
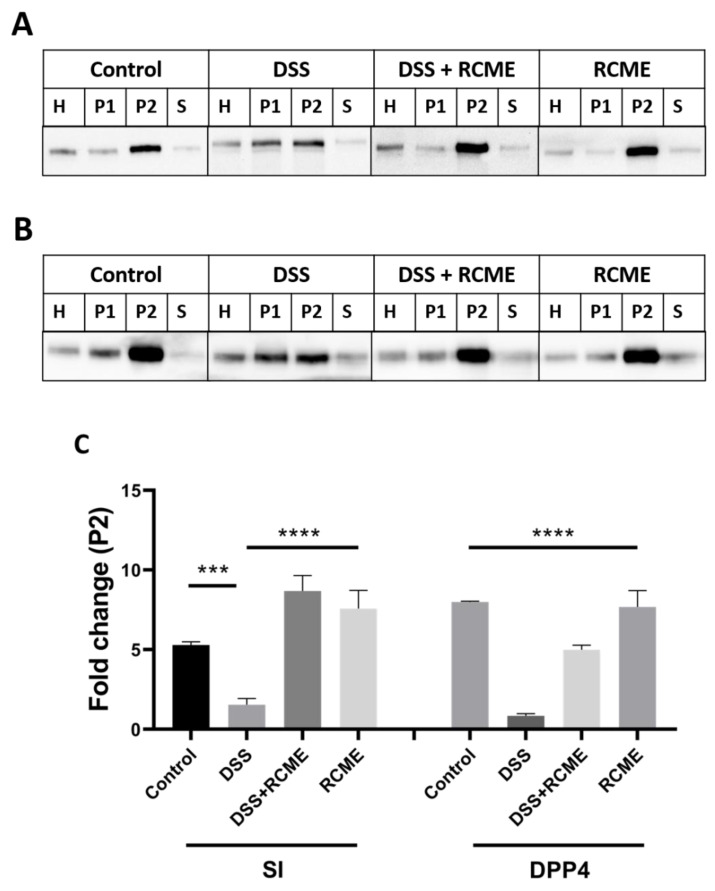
*Rosa canina* methanol extract (RCME) treatment improves dextran sulfate sodium (DSS)-impaired sorting of sucrase-isomaltase (SI) and dipeptidyl peptidase-4 (DPP4). Caco-2 cells were treated or non-treated with DSS, DSS and RCME or RCME. 24 h post-treatment, the cellular homogenates (H) were fractionated into intracellular and basolateral membranes (P1), brush border membranes (P2) and soluble and vesicular fraction (S). The fractions were subjected to SDS-PAGE for SI (**A**) and DPP4 (**B**) analysis. In (**C**), SI or DPP4 in P2 from the different preparations are compared. Tukey’s multiple test, *** *p* < 0.001, **** *p* < 0.0001 versus DSS-treated group, S.E.M., *n* = 3.

**Figure 5 nutrients-13-00441-f005:**
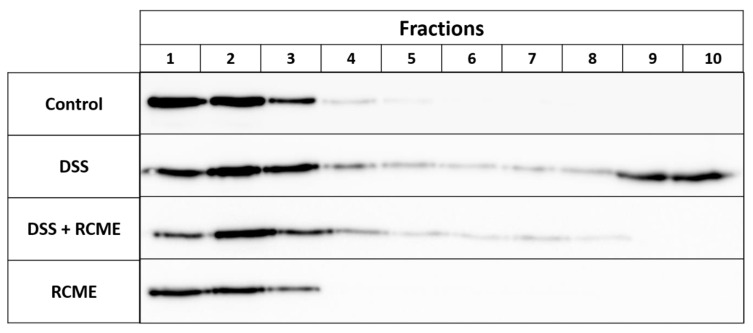
Dextran sulfate sodium (DSS)-induced lipid rafts (LRs) alteration was recovered upon *Rosa canina* methanol extract (RCME) Figure 2. cells were treated or non-treated with DSS, DSS and RCME or RCME. 24 h post-treatment, the cells were lysed with Triton X-100 and the LRs were isolated on sucrose density gradient. The 10 collected fractions were subjected to SDS-PAGE for flotillin-2 (FLOT2) analysis.

**Figure 6 nutrients-13-00441-f006:**
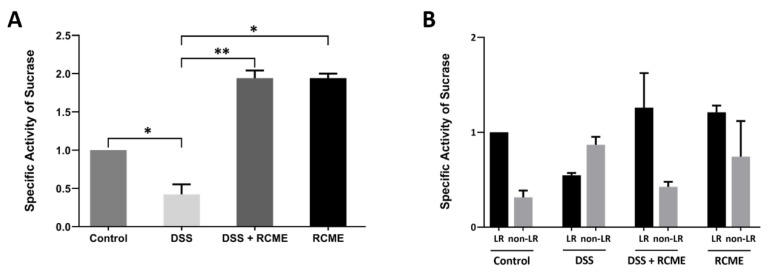
*Rosa canina* methanol extract (RCME) treatment restores and improves dextran sulfate sodium (DSS)-induced reduced specific activity of sucrase-isomaltase (SI). Caco-2 cells were treated or non-treated (control) with DSS, DSS and RCME or RCME. 24 h post-treatment, the relative activity of SI in cellular homogenate (**A**) or in lipid rafts (LRs)/non-lipid rafts (non-LRs) fractions (**B**) was measured versus sucrose as a substrate. The specific activity was assessed by correlating the measured activity with the band intensity that was visualized by Western blotting. Tukey’s multiple test, * *p* < 0.05, ** *p* < 0.01 versus DSS-treated group, S.E.M., *n* = 3.

**Table 1 nutrients-13-00441-t001:** List of primers for ER stress markers.

Primer	Sequence
ATF4 fw ATF4 rev	GTTCTCCAGCGACAAGGCTA ATCCTGCTTGCTGTTGTTGG
BiP fw BiP rev	TGTTCAACCAATTATCAGCAAACTC TTCTGCTGTATCCTCTTCACCAGT
CHOP fw CHOP rev	AGAACCAGGAAACGGAAACAGA TCTCCTTCATGCGCTGCTTT
EDEM fw EDEM rev	CAAGTGTGGGTACGCCACG AAAGAAGCTCTCCATCCGGTC
Grp94 fw Grp94 rev	GAAACGGATGCCTGGTGG GCCCCTTCTTCCTGGGTC
XBP1 fw XBP1 rev	TGGCCGGGTCTGCTGAGTCCG ATCCATGGGGAGATGTTCTGG
GAPDH Fw GAPDH rev	CATGGCCTTCCGTGTTCCTA CCTGCTTCACCACCTTCTTGAT

fw: forward; rev: reverse; ATF4: activation transcription factor 4; BiP: immunoglobulin-binding protein; CHOP: C/EBP homologous protein; EDEM: ERAD-enhancing α-mannosidase-like proteins; Grp94: glucose regulated protein 94; XBP1: X-box binding protein; GAPDH: Glyceraldehyde 3-phosphate dehydrogenase.

## Data Availability

Data available in a publicly accessible repository.

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
