# Peer review of "Rosa canina L. Can Restore Endoplasmic Reticulum Alterations, Protein Trafficking and Membrane Integrity in a Dextran Sulfate Sodium-Induced Inflammatory Bowel Disease Phenotype"

_nutrients, 2021, doi:10.3390/nu13020441_

Round 1
Reviewer 1 Report
Abstract
Line 16 polyphenol-rich
Line 23 This is a run-on sentence and its meaning is not clear.
The abstract could be improved with a little more detail to what was actually done in the study and more conclusive statements about the findings overall. The level of detail attributed to the results overwhelms without sufficient information about the study (i.e. treatment levels, time of administration in comparison to DSS, etc.).
Introduction
Really nicely written, provides a solid background for the study.
Methods
How was confluency determined? Visually or with a quantitative measure?
What passages were used for the experiments?
Were any chemical analyses of the extract carried out, i.e. total polyphenols, any kind of characterization? While maybe not germane to this study, this information would be useful to anyone else investigating this treatment in the future.
There is some very basic information about how the cells were treated that is missing from the methods section that makes it very unclear. Please clarify the treatment groups. The statement of “incubated for 24 h with DSS and/or RCME” suggests that cells were exposed to DSS, RCME and RCME+DSS, but the concentrations of RCME listed are not consistent wit the 100 ug/ml listed in the following paragraph. It all makes sense once you reach Fig 2, but it really must be clarified in the text beforehand.
Where methods are described as being “performed according to”, it would be helpful to add a short description anyway.
Why was SEM used instead of SD?
Please specify the instances in which more than three independent experiments were used.
Results/Discussion
Figure 4 would be better with some reorganization of the data. The bands look nice, but the graph should be rearranged to compare all of the P1, all of the P2, all of the S so you don’t have the huge bars for statistical significance. If you broke them out even further onto their own graphs, we could get a better idea of the non-significant differences between the P1 and S groups. Since you aren’t comparing P1 v P2 v S anyway, it doesn’t really make sense to have them on the same graphs.
Overall, I think this is a decent paper with a couple significant, but fixable flaws in writing and presentation.
Author Response
Responses to Reviewer # 1
We thank the reviewer for the positive and constructive assessment of our manuscript. All the comments raised have been addressed in the revised version of the manuscript. We should indicate that we have divided the Results/Discussion section into two separate sections along the suggestion made by reviewer # 2. Because of this change in the format of the manuscript the tracked changes will confuse the reader. Therefore, we have marked the changes suggested in yellow.
In what follows are our responses:
- Abstract
Reviewer:
- Line 16: polyphenol-rich;
- Line 23 This is a run-on sentence and its meaning is not clear.
- The abstract could be improved with a little more detail to what was actually done in the study and more conclusive statements about the findings overall. The level of detail attributed to the results overwhelms without sufficient information about the study (i.e. treatment levels, time of administration in comparison to DSS, etc.).
Response:
We made the corrections in lines 16 and 23 as suggested and more details have been provided in the abstract.
- Introduction
Reviewer:
- Really nicely written, provides a solid background for the study.
Response:
We are pleased to read this positive assessment.
- Methods
Reviewer:
- How was confluency determined? Visually or with a quantitative measure?
- What passages were used for the experiments?
Response:
The Caco-2 cells confluence was assessed by microscopy. The cells were examined daily and when confluence was reached (day 0) the cells were kept for 6 more days, followed by treatments or non-treatments for 24 hours. The cells were used at passages 20-25.
Reviewer:
- Were any chemical analyses of the extract carried out, i.e. total polyphenols, any kind of characterization? While maybe not germane to this study, this information would be useful to anyone else investigating this treatment in the future.
Response:
Indeed, the chemical-analysis of Rosa canina methanol extract was performed using Liquid Chromatography-Electrospray Ionization-Tandem Mass Spectrometry (LC-ESI-MS) in previous studies. The paper describing this work is cited in this manuscript in reference number 16 “Chemical Characterization of Bioactive Components of Rosa canina Extract and Its Protective Effect on Dextran Sulfate Sodium-Induced Intestinal Bowel Disease in a Mouse Model, by Wanes et al.”
Reviewer:
- There is some very basic information about how the cells were treated that is missing from the methods section that makes it very unclear. Please clarify the treatment groups. The statement of “incubated for 24 h with DSS and/or RCME” suggests that cells were exposed to DSS, RCME and RCME+DSS, but the concentrations of RCME listed are not consistent wit the 100 ug/ml listed in the following paragraph. It all makes sense once you reach Fig 2, but it really must be clarified in the text beforehand.
Response:
We fully agree that more details are needed for the sake of clarity. We have edited this part and provided these basic information as suggested.
Reviewer:
- Where methods are described as being “performed according to”, it would be helpful to add a short description anyway.
Response:
The Methods section contains now substantially more details than before.
Reviewer:
- Why was SEM used instead of SD?
Response:
Our results were expressed as mean ± SEM in order to indicate the accuracy of the mean. In other word, the SEM is an estimate of variability of possible values of means of samples. So we used SEM to show the precision of the sample mean which represents precisely the entire group and precision of the study.
Reviewer:
- Please specify the instances in which more than three independent experiments were used.
Response:
All the experiments described in this manuscript have been performed at least three times. In Fig. 3 this was not indicated. We have added it now.
- Results/Discussion
Reviewer:
- Figure 4 would be better with some reorganization of the data. The bands look nice, but the graph should be rearranged to compare all of the P1, all of the P2, all of the S so you don’t have the huge bars for statistical significance. If you broke them out even further onto their own graphs, we could get a better idea of the non-significant differences between the P1 and S groups. Since you aren’t comparing P1 v P2 v S anyway, it doesn’t really make sense to have them on the same graphs.
Response:
We thank the reviewer for this nice suggestion, which is implemented in revised Fig. 4. Along this, we have added anothr part “C” to A and B in Fig. 4, in which P2 fractions from all samples treartemnts for SI and DPP4 are compared to each other. This makes indeed the presentation much more clearer and better for the reader to comprehend.

Reviewer 2 Report
In the present basic science article Wanes et al demonstrated that a methanol extract of Rosa canina (RCME), which contains several anti-oxidant agents, may reverse in mice chemical colitis induced by DSS. It was shown that this effect was mediated by a restoration of altered membrane trafficking of sucrose isomaltase and DPP4.
The experiments are interesting and well performed, however the text is badly organized. I do not understand why results and discussion are merged in the same paragraphs. This makes the paper confusing and hard to read. Please re-write.
Author Response
Responses to Reviewer # 2
- In the present basic science article Wanes et al demonstrated that a methanol extract of Rosa canina (RCME), which contains several anti-oxidant agents, may reverse in mice chemical colitis induced by DSS. It was shown that this effect was mediated by a restoration of altered membrane trafficking of sucrose isomaltase and DPP4.
The experiments are interesting and well performed, however the text is badly organized. I do not understand why results and discussion are merged in the same paragraphs. This makes the paper confusing and hard to read. Please re-write.
Response:
We thank the reviewer for the overall positive and constructive assessment of our manuscript and also for the suggestion to divide the Results/Discussion section into two separate sections. We have re-written these parts and we agree that with this restructuring the data are more clearly presented and discussed.

Round 2
Reviewer 2 Report
The paper may be accepted